# Social Connectedness and Associations with Gambling Risk in New Zealand

**DOI:** 10.3390/jcm11237123

**Published:** 2022-11-30

**Authors:** Grace Y. Wang, Maria E. Bellringer

**Affiliations:** 1School of Psychology and Wellbeing, University of Southern Queensland, Toowoomba, QLD 4350, Australia; 2Centre for Health Research, University of Southern Queensland, Toowoomba, QLD 4350, Australia; 3Gambling and Addictions Research Centre, Auckland University of Technology, Auckland 1010, New Zealand

**Keywords:** social connectedness, gambling, leisure activities, indigenous population

## Abstract

Multiple factors are associated with disordered gambling, with some populations having a greater risk for developing disordered gambling than others. The present study, utilising data previously collected for a New Zealand (NZ) national gambling survey, explored the associations of social connectedness and leisure activities with risky gambling behaviour and quality of life. Poorer social connectedness and leisure activities were found to be associated with increased gambling risk and poorer quality of life, respectively. Social connectedness and leisure activities strongly predicted type of gambling activities and quality of life. Furthermore, Māori (NZ’s indigenous population) had lower social connectedness and fewer leisure activities, and a greater gambling risk, as well as higher psychological distress, than the NZ European/Other population. These findings indicate that the risk of progressing from recreational gambling to risky gambling is relatively higher for Māori, and that social connectedness and leisure activities could be contributing factors for this increased risk. It is, therefore, important that social connectedness and leisure activities are seriously considered in public health and treatment efforts to reduce gambling harm for vulnerable populations.

## 1. Introduction

Harmful gambling refers to gambling behaviours which may not meet the diagnostic criteria of gambling disorder, but which lead to significant harm to individuals and communities. Harmful gambling is commonly framed as a public health issue, particularly by countries such as Australia [1], New Zealand (NZ) [2], Canada [3], Sweden [4] and the United Kingdom [5]. However, there is an absence of consensus in theories to explain why some people develop harmful gambling behaviours while others do not.

Cognitive Theory suggests that gambling disorder is caused by false belief and assumptions about an individual’s skills and randomness of gambling winnings [6,7]. Behaviour Theory considers gambling behaviour is learnt and reinforced for repetitive occurrences through winning rewards [8]. Biopsychosocial Theory, considered a more comprehensive model, argues that gambling behaviour represents a complex and multifaceted phenomenon that is linked to a combination of biological, psychological, and social factors, such as impulsivity, early exposure to gambling and social support [9,10]. The Pathways Model posits that disordered gambling behaviour is a complex combination of determinants and that gamblers are a heterogeneous group [11]. This theory suggests there are three subgroups of gamblers that can be identified: (1) behaviourally conditioned, (2) emotionally vulnerable and (3) antisocial impulsivist gamblers [11].

Over the years, a large volume of research has been conducted to determine risk factors associated with gambling disorders. Higher levels of problematic gambling have been noted amongst indigenous and migrant groups in many countries. For example, African Americans in the United States of America have been reported to have a higher prevalence of problem gambling compared with Caucasians [12]. Similarly, increased problematic gambling has been observed amongst the Canadian indigenous population compared with the non-indigenous population [13] and with the indigenous communities of Australia compared to the non-indigenous communities [14]. In NZ, Māori (NZ’s indigenous population) and Pacific peoples are more likely than NZ European/Others to be moderate-risk/problem gamblers [15,16]. In 2020, for example, 3.7% of Māori and 3.0% of Pacific people were classified as problem or moderate risk gamblers compared with 1.4% of European/Other/Others [17]. Furthermore, casinos pose particular risk for Asian groups [18].

Although cultural differences may play a role, influencing individual variations in gambling behaviours [19,20,21], it has been argued that race and ethnic minority status themselves are not a risk factor for gambling disorder but underlying potential risk factors related to this status are. For example, experiences of social exclusion, the stress of acculturation and disadvantaged neighbourhoods [22,23]. Research has shown that people may use gambling as a way of escape from negative emotional states such as depression or stress, for enjoyment or excitement from an adrenalin-driven activity, or for social aspects [24,25]. Furthermore, in many countries including NZ, gambling availability is disproportionately higher in areas of lower socio-economic status, exacerbating the risk to disadvantaged populations living in such areas [26,27,28]; In other words, the gambling environment and gambling activity may provide an alternative social networking platform, facilitating human interaction and leading to a sense of belonging for those who are socially isolated [29,30].

Disordered gamblers typically engage in few social and recreational activities apart from gambling [31] and major relapses commonly occur when gamblers are alone, facing a non-gambling life on their own [32]. Social connectedness, considered a psychological sense of belonging to a group and interpersonal closeness with society, has generally been shown to be beneficial, promoting individual well-being, reducing the risk of developing addictive behaviours, and facilitating recovery from addictions [33,34]. Leisure activities help to decrease daily stress and tension and are crucial for individuals’ social development, including building social relationships, and acquiring additional skills and knowledge [35]. Unfortunately, ethnic minorities or subgroups often report high levels of social isolation and loneliness [36,37,38]. In NZ, Māori kaumātua (elders) experience varying degrees of cultural dissonance and have reported feelings of separation and social isolation [39], while Pacific people have reported generally being well connected socially with others but with a significantly lower perceived social support than Māori and other groups (i.e., non-Māori, non-Pacific people) [40].

While associations between social connectedness and gambling are well supported by evidence, it is unclear whether social connectedness could explain individual progression to disordered gambling. The aim of this study was to explore the associations between social connectedness, leisure activities and gambling risk (including frequency and type of gambling activities) and quality of life. Improving understanding of their interactions could contribute to the development of a comprehensive explanatory model of gambling behaviour, informing improved gambling treatments and public health approaches to reduce gambling harm. It was hypothesised that social connectedness and leisure activities would be positively associated with gambling risk and influence individuals’ quality of life.

## 2. Methods

### 2.1. Participants

This study involved secondary analysis of data collected from the baseline wave (in the year 2012) of the population representative NZ National Gambling Study. Ethical approval was granted by the Northern Y Regional Ethics Committee of the Health and Disability Ethics Committees on 26 May 2011 (Reference: NTY/11/04/040). Although the baseline data comprised 6251 participants, the current analysis only included data from the 4904 participants who reported involvement in at least one gambling activity in the prior 12 months, ethnicity and completed all primary measures stated below. The data from other 1284 participants were excluded.

### 2.2. Procedure

The design and methods of the NZ National Gambling Study have previously been published [18]. In brief, this was a nationwide survey of adults aged 18 years and older who lived in a private dwelling. A stratified three-stage cluster design was used with the three strata being: district health board regions, census mesh blocks, and private dwellings. Recruitment and interviewing were both face-to-face, with participants interviewed in their homes via Computer Assisted Personal Interviews (CAPI). The response rate was 64%. In order that findings could be generalised to the NZ population, weightings were applied to account for selection probability, and to adjust for demographics (age, gender and ethnicity) relative to census proportions. However, for the current study, raw values rather than weighted values were used.

### 2.3. Measures

#### 2.3.1. Gambling Risk Level

The nine-item Problem Gambling Severity Index (PGSI) [41] was used to measure gambling risk level in a past 12 month time frame. Each PGSI item had four response choices: ‘never’, ‘sometimes’, ‘most of the time’, or ‘almost always’. The total score ranged from 0 to 27. Gambling risk level is based on the score; non-gambler, non-problem gambler (score 0), low risk gambler (score 1 or 2), moderate risk gambler (score 3 to 7), or problem gambler (score 8 to 27). For this study, PGSI scores were used as a continuous variable. The PGSI has a high internal reliability (minimum Cronbach’s alpha of 0.86) and has been shown to be robust and reliable in the NZ population [42].

#### 2.3.2. Gambling Frequency and Participation in Specific Gambling Activities

An 11-point rating scale was used to assess participants’ overall frequency of gambling, starting with “1” representing four times a week to “11” representing less frequently than once a year. Scores were reversed for analysis. Participation in specific gambling activities over the last 12 months, such as on casino table games, electronic gaming machines (EGMs), lottery tickets and so on, was also assessed. Each item had two response choices: “Yes” or “No”. Higher scores indicate participation in a larger number of gambling activities.

#### 2.3.3. General Psychological Distress

The Kessler-10 (K-10) questionnaire was included to provide a continuous measure of general psychological distress that is responsive to change over time. The K-10 has been well validated internationally and was included due to its brevity and simple response format. It also produces a summary measure indicating probability of currently experiencing an anxiety or depressive disorder [43].

#### 2.3.4. Quality of Life

Quality of life was assessed by the WHOQoL-8, an eight-item version of the widely used 26-item WHOQoL-Bref. This short form has been used in a number of countries, is robust psychometrically, and overall performance is strongly correlated with scores from the original WHOQoL instrument [44].

### 2.4. Leisure Activity

Level of leisure activity involvement was assessed by ‘Yes’ or ‘No’ responses to a list of 20 common individual and social leisure activities such as reading for pleasure, spending time with friends/family, and being involved in voluntary community work. An ‘other’ category was also included. Higher scores indicated involvement in a higher number of leisure activities.

#### 2.4.1. Social Connectedness

Individual questions on access to help (“Can you get help from family, friends or neighbours when you need it?”), community involvement (“Are you a member of an organised group such as a sports or church group or another community group including those over the internet?” and “Do you like living in your community?”) and quality of community services (“How would you rate the overall quality of services, facilities and ‘things to do’ in your community?”) based on those used in the Victorian Gambling Study [45] were administered to assess individuals’ social connectedness. Higher scores indicated a greater level of social connectedness.

#### 2.4.2. Demographics

Data on age, gender, ethnicity, education, and employment status were collected.

### 2.5. Data Analysis

First, descriptive analysis was conducted. Differences in gambling risk, quality of life, psychological distress, social connectedness, and number of leisure activities between ethnic groups were explored using Kruskal–Wallis test. Significant effects were followed-up with post hoc tests for multiple comparisons adjusted using Bonferroni correction. Sex difference in gambling and quality of life, psychological distress, social connectedness, and number of leisure activities was also explored for the whole sample, using Mann–Whitney U test. Associations between social connectedness, leisure activities, psychological distress and PGSI, were explored using Spearman’s correlation. Poisson regression was employed to examine the overall frequency of gambling (from “four times a week or more”, to “Less frequently than once a year”) and number of gambling activities based on predictors of social connectedness, psychological distress, number of leisure activities, and ethnicity, respectively. Multiple regressions were performed to determine the relative effects of social connectedness, leisure activities, and psychological stress and gambling risk, on quality of life. A two-tailed level of significance was set to *p* < 0.05 before correction. Statistical analyses were performed in IBM SPSS Statistics (version 28).

## 3. Results

### 3.1. Participants

Of the total 4904 gamblers, mean age was 47.9 ± 17.0 years (range 18 to 93 years), 20.3% self-identified as Māori (*n* = 997), 11.9% self-identified as Pacific (*n* = 582), 10.1% reported Asian (*n* = 493) identity, and 57.7% were NZ European/Other (*n* = 2832).

### 3.2. Comparison of Gambling Risk, Quality of Life, Psychological Distress, and Social Connectedness between Ethnic Groups and Sex

Table 1 shows demographic features and participant-reported outcome across ethnic groups. There were significant group differences in gambling risk (PGSI scores) (H = 147.87, *p* < 0.001), psychological distress (H = 58.4, *p* = 0.009), social connectedness (H = 80.1, *p* < 0.001), leisure activities (H = 21.5, *p* < 0.001) and quality of life (H = 32.3, *p* < 0.001). Post hoc tests show that Māori participants had higher PGSI and psychological distress scores, lower social connectedness and leisure activity involvement, and poorer quality of life relative to NZ European/Other participants. Similarly, Pacific participants had higher PGSI and psychological distress scores, and poorer quality of life relative to the NZ European/Other group. However, social connectedness and leisure activity involvement did not differ between Pacific and NZ European/Other groups. Asian participants had higher PGSI scores, lower social connectedness, and poorer quality of life relative to the NZ European/Other group, but the number of leisure activities and level of psychological distress between the two groups was not different (Figure 1).

Furthermore, male participants exhibited greater gambling risk (U = −2.496, *p* = 0.013), psychological distress (U = −5.556, *p* < 0.001) and less involvement of leisure activities (U = 4.419, *p* < 0.001), but they were not different from female participants in terms of quality of life.

### 3.3. Correlations of Gambling Risk with Quality of Life, Psychological Distress, Social Connectedness, Leisure Activities, and Age

Social connectedness was significantly correlated with gambling risk (r = −0.07, *p* < 0.001), quality of life (r = 0.24, *p* < 0.001), psychological distress (r = −0.11, *p* < 0.001) and leisure activities (r = 0.22, *p* < 0.001). Furthermore, gambling risk was significantly correlated with quality of life (r = −0.05, *p* < 0.001), psychological distress (r = 0.19, *p* < 0.001), and leisure activities (r = −0.03, *p* = 0.03). Age was negatively correlated with gambling risk (r = −0.122, *p* < 0.001), psychological distress (r = −0.156, *p* < 0.001) but positive correlated with quality of life (r = 0.043, *p* < 0.001) and social connectedness (r = 0.162, *p* < 0.001) (Table 2).

### 3.4. Predicting Gambling Risk by Ethnicity, Social Connectiveness, Leisure Activities and Psychological Distress

A Poisson regression was run to predict the overall frequency of gambling, based on ethnicity, social connectedness, psychological distress, and leisure activities (Table 3). Goodness of Fit showed the value is more than 0.05, indicating the model fits the data well. Direct effects were significant, omnibus χ^2^ = 121.80, df = 6, *p* < 0.001. Ethnicity and number of leisure activities significantly predicted frequency of gambling, whereas social connectedness and psychological distress did not predict gambling frequency. For every additional leisure activity involvement, frequency for gambling participation was 0.991 times reduced (95% CI, 0.998 to 0.994), *p* < 0.001. Furthermore, compared to NZ European/Other participants, gambling frequency increased for Māori and Pacific participants, 1.09 (95% CI, 1.057 to 1.119) and 1.07 (95% CI, 1.036 to 1.112) times, respectively, *p* < 0.001, whereas for Asian participants, frequency of gambling decreased 0.895 times (95% CI, 0.859 to 0.933), *p* < 0.00.

For prediction of gambling activity count, Goodness of Fit showed the value is more than 0.05, indicating the model fits the data well. Direct effects were significant, omnibus χ^2^ = 32.91, df = 6, *p* < 0.001. Social connectedness and number of leisure activities significantly predicted gambling activity involvement count, while psychological distress did not predict gambling activity count. For every extra score of social connectedness, 0.992 times (95% CI, 0.984 to 1) gambling activity involvement count decreased, *p* = 0.04, while gambling activity involvement count increased 1.01 times (95% CI, 1.005 to 1.020) with every additional leisure activity. Furthermore, compared to NZ European/Other participants, Māori had 1.14 times (95% CI, 1.074 to 1.207) risk for increased gambling activity count, *p* < 0.001. No ethnicity effect in predicting gambling activity was observed in other ethnic groups relative to NZ European/Other participants (Table 4).

### 3.5. Prediction of Quality of Life by Social Connectiveness, Leisure Activity, Psychological Distress and Gambling Risk

Multiple regression was performed to ascertain the effects of social connectiveness, number of leisure activities, psychological distress, and gambling risk on quality of life. The model was statistically significant, showing that social connectedness, β = 0.28, *p* < 0.001, 95% CI [0.244, 0.323], number of leisure activities, β = 6.86, *p* < 0.001, 95% CI [0.08, 0.158], psychological distress, β = −35.91, *p* < 0.001, 95% CI [−0.454, −0.407], and PGSI scores, β = −2.74, *p* < 0.001, 95% CI [−0.175, −0.029], were significant predictors, explaining a significant amount of the variance in quality of life, *R*^2^  =  0.29, *F* (4, 4877)  =  489.67, *p* < 0.001 (Table 5).

## 4. Discussion

Evidence suggests that there is no single cause of disordered gambling yet understanding the factors that influence transitions from recreational to risky levels of gambling are important for harm prevention. The present study utilised an available national gambling survey dataset to explore the role of social connectedness and leisure activities in gambling risk and their association with psychological distress and quality of life. Consistent with the previous findings which argue the important role of social connection in prevention of gambling and health problem [46,47,48], We found that poorer social connectedness was associated with increased gambling risk, greater psychological distress, fewer leisure activities and poorer quality of life. Stronger social connections were associated with fewer gambling activities and better quality of life. Evidence suggests that strong social relationships could function as a buffer against gambling problems, and persons with poor social connectedness engage in gambling for a social purpose may not necessarily receive social benefits for doing it [49]. In line with this, our findings implicate the protective effect of social connectedness against problem gambling. This is probably particularly relevant to male and younger population given their greater risk in gambling which was indicated by previous research [50] and current findings. However, neither social connectedness nor psychological distress were significant predictors of gambling frequency. In contrast, greater leisure activity participation was associated with reduced gambling frequency.

Social connectedness, e.g., personal bonds with others and communities, is an essential part of being human, leading to a sense of well-being, belonging and cohesion [34,51]. In contrast, social isolation and disconnectedness not only cause feelings of alienation but also psychological distress [52]. The need to connect with others has been frequently identified as a motivational factor for gambling [53]. Research shows that both recreational and disordered gamblers could perceive gambling as an acceptable leisure activity and be attracted to it for socialisation, escaping from daily problems [34], or to cope with negative emotions of social exclusion [54,55]. Although severity of psychosocial problems varies by gambling activity and certain activities are more risky than others (e.g., EGMs), gambling on multiple activities is often problematic [56]. Engaging in multiple gambling activities could be an indication of attachment to the fundamental essence of the gambling experience [57]. Our findings implicate that weaker social connectedness is associated with increased risky gambling, and this association is related to participation in a greater number of gambling activities.

Furthermore, indigenous and ethnic minority groups have long been known to have disproportionately higher risk for developing disordered gambling behaviours and associated with this are poorer mental well-being and other addictions [22,58,59]. In our study, Māori and Pacific participants reported greater frequency of gambling participation and increased involvement in multiple activities compared with NZ European/Other participants. Furthermore, participants in our ethnic sub-groups (Māori, Pacific and Asian), all had a higher score on the PGSI, and reported poorer quality of life than participants who identified as NZ European/Other, although Asian participants reported reduced gambling frequency. It has been argued that prevalence of disordered gambling varies within and across indigenous populations due to differences in their history and involvement with gambling [59,60]. It is also partly related to inequitable distribution of gambling venues and availability in low socio-economic areas, where indigenous and migrant populations often reside [61]. Personal wellbeing has been related to cultural wellbeing in indigenous and other populations [62]. Thus, gambling participation, including its benefits and harms can affect culture in distinctive yet complex ways [16,63]. Taken together, our findings suggest that Māori and Pacific people in NZ have a higher risk for developing disordered gambling partly related to weaker social connectedness.

Research shows that leisure activities explain a significant part of individuals’ social connectedness [64], which is also positively associated with mental well-being and negatively associated with depression/anxiety symptoms [65]. In the present study, we found that leisure activities were associated with social connectedness, and modulated frequency of gambling and number of gambling activities participated in. Greater involvement in leisure activities, such as taking part in sports or listening to/playing music, is associated with a reduction in gambling frequency but an increase in number of gambling activities participated in. Although the reasons for these findings are unclear, it is possible that most of our participants were non-problem gamblers, whose motivation for gambling could have been less related to winning money but rather perceived as a normal leisure activity. Thus, a person with more leisure activities may wish to try different gambling activities, at a low frequency, for the purpose of relaxation. Although research shows that among socialisation, amusement, avoidance, excitement, and monetary motives, the latter is the only factor showing a direct positive influence on severity of gambling [25], the increased risk associated with participation in a higher number of gambling activities cannot be ignored. Taken together, our findings suggest that the effect of leisure activities on gambling is more complicated than expected and can be either positive or negative. Future research to understand this aspect is required.

The limitations of this study need to be acknowledged. First, we only adapted some of the survey items measuring social connectedness and this is likely to have reduced power to capture the whole picture of a person’s social life. Furthermore, our findings did not consider the fact that a person’s social connectedness might fluctuate over time. Finally, raw numbers and not weighted data were used meaning that the findings from this analysis cannot necessarily be generalised to the whole NZ population. Nevertheless, our findings provide a preliminary snapshot of the relationship between social connectedness and gambling in the NZ context. In addition, diversity in ethnic identity should be acknowledged as ethnicity can be either obvious or ambiguous and a person could belong to more than one ethnic group.

In conclusion, our findings highlight that the effects of risk factors for disordered gambling vary with ethnicity, as well as the importance of addressing social connectedness and mental well-being in reducing harmful effects of gambling participation. When a country consists of diverse ethnicities, as in the case of NZ which comprises a significant proportion of indigenous people (Māori), a rapidly growing Asian population and a distinct Pacific population, gambling research and policy development should take ethnic differences into consideration.

## Figures and Tables

**Figure 1 jcm-11-07123-f001:**
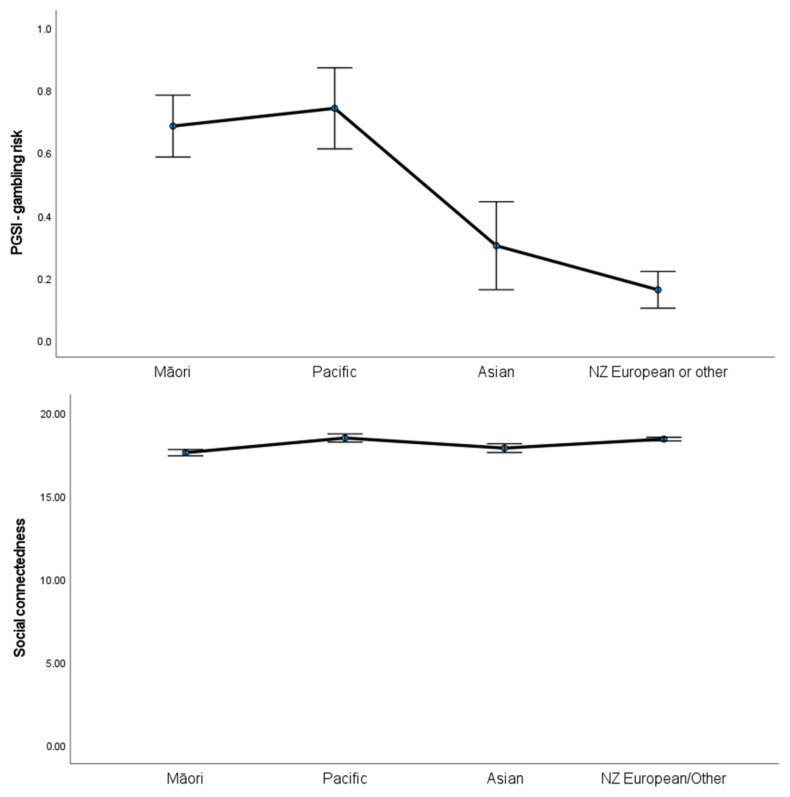
Mean and 90% confidence interval of gambling risk, social connectedness, leisure activities, psychological distress, and quality of life among groups.

**Table 1 jcm-11-07123-t001:** Demographic and outcome measures across groups.

	Māori	Pacific	Asian	NZ European/Other
	(*n* = 997)	(*n* = 582)	(*n* = 493)	(*n* = 2832)
	Mean (SD)	Mean (SD)	Mean (SD)	Mean (SD)
Age (years)	43.03 (15.31)	41.41 (13.78)	40.27(13.13)	52.33 (17.45)
Female (%)	62.7	59.3	50.1	56.1
Highest Education (%)				
No formal Qual	30.0	19.6	4.7	17.2
Secondary School	21.9	35.9	19.9	21.4
Vocational or Trade	23.6	22.3	10.5	24.2
University Degree or Higher	24.4 *	22.2	64.9	37.2
Employment (%)				
Full time	43.0	50.5	60.6	44.4
Par time	15.4	10.5	12.2	17.6
Currently search for job	7.2	7.4	4.7	2.6
Student	4.3	4.6	6.5	2.1
Homemaker	8.7	10.3	8.3	5.5
Beneficiary	13.2	9.6	1.0	4.4
Retired	7.4	6.0	6.3	22.8
Other	0.6	1.0	0.4	0.5
Social connectedness	17.65 (3.07)	18.50 (2.97)	17.89 (2.87)	18.43 (3.00)
Leisure	8.62 (3.49)	8.96 (3.81)	8.97 (3.49)	9.09 (3.27)
Quality of life	23.81 (5.22)	24.16 (4.81)	24.43 (4.45)	24.83 (4.67)
Psychological distress	5.34 (6.13)	5.29 (5.82)	3.84 (4.69)	3.77 (4.41)
PGSI score	0.68 (2.48)	0.74 (2.25)	0.30 (1.14)	0.16 (0.97)

PGSI: Problem Gambling Severity Index; * One subject did not report education.

**Table 2 jcm-11-07123-t002:** Spearman’s correlations of social connectedness, leisure activity, quality of life and gambling.

	1	2	3	4	5	6
1. Social connectedness	-					
2. Leisure activity	0.220 **	-				
3. Quality of life	0.244 **	0.123 **	-			
4. Psychological distress	−0.114 **	0.023	−0.440 **	-		
5. PGSI score	−0.068 **	−0.032 *	−0.151 **	0.192 **	-	
6. Age	0.162 **	−0.088 **	0.043 **	−0.156 **	−0.122 **	-

**. Correlation is significant at the 0.01 level (2-tailed); *. Correlation is significant at the 0.05 level (2-tailed).

**Table 3 jcm-11-07123-t003:** Poisson regression results—frequency of gambling count, social connectedness, leisure activities, psychological distress, and ethnicity.

Variables	B	SD	*p*	Exp(B)	95% Confidence Interval for Exp(B)	
					Lower	Upper
Social connectedness	−0.001	0.0020	0.547	0.999	0.995	1.003
Leisure activities	−0.009	0.0018	<0.001	0.991	0.988	0.994
Psychological distress	−0.002	0.0012	0.092	0.998	0.996	1.000
Māori ^a^	0.084	0.0147	<0.001	1.088	1.057	1.119
Pacific ^a^	0.071	0.0181	<0.001	1.074	1.036	1.112
Asian ^a^	−0.111	0.0209	<0.001	0.895	0.859	0.933

B: Unstandardised coefficient; SD: Standard error; *p*: hypothesis test significance value; Exp(B): exponentiated regression coefficient; ^a^: Reference category is NZ European/Other group.

**Table 4 jcm-11-07123-t004:** Poisson regression results—gambling activity count, social connectedness, leisure activities, psychological distress, and ethnicity.

Variables	B	SD	*p*	Exp(B)	95% Confidence Interval for Exp(B)	
					Lower	Upper
Social connectedness	−0.008	0.004	0.043	0.992	0.984	1.000
Leisure activities	0.012	0.004	<0.001	1.012	1.005	1.020
Psychological distress	−0.003	0.0024	0.276	0.997	0.993	1.002
Māori ^a^	0.130	0.0298	<0.001	1.139	1.074	1.207
Pacific ^a^	0.063	0.0374	0.092	1.065	0.990	1.146
Asian ^a^	0.010	0.0406	0.809	1.010	0.933	1.094

B: Unstandardised coefficient; SD: Standard error; *p*: hypothesis test significance value; Exp(B): exponentiated regression coefficient; ^a^: Reference category is NZ European/Other group.

**Table 5 jcm-11-07123-t005:** Regression analysis for quality-of-life prediction.

Predictor	B	SD	t	*p*
Social connectedness	0.283	0.020	14.185	<0.001
Leisure activities	0.123	0.018	6.862	<0.001
Psychological distress	−0.430	0.012	−35.911	<0.001
PGSI score	−0.102	0.037	−2.743	0.006

B: Unstandardised coefficient; SD: Standard error.

## Data Availability

Data available on request from the author.

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
