# Peer review of "Social Connectedness and Associations with Gambling Risk in New Zealand"

_jcm, 2022, doi:10.3390/jcm11237123_

Round 1

Reviewer 1 Report

Thanks for the Editor's invitation and the authors' hard work in this field.  This paper is a secondary data analysis to explore the associations of social connectedness and leisure activities with risky gambling behaviour and quality of life. This study contributes to the evidence required to increase the social connectedness and leisure activities in public health and treatment efforts. Generally, the quality of this manuscript needs to be further improved with appropriate revisions. In particular, some concerns needed to be addressed.

My comments are listed below:

1.       Please provide further details for the NZ National Gambling Study. Did the 4,904 participants complete all six scores?

2. Whether age and gender are related to gambling risk? would the age and gender difference affect the final results

3. From table 1, demographic data is too simple, further details data is required. And Figure is more intuitive for data presentation, such as PGSI score.

4. Further discussions for the differences and similarities between the present results and other studies are supposed to be added.

5. Line 290 and 293: The format of references in the manuscript is not uniform.

Author Response

  1. Please provide further details for the NZ National Gambling Study. Did the 4,904 participants complete all six scores?

Response: yes, they did. We have added this in the main text.

  1. Whether age and gender are related to gambling risk? would the age and gender difference affect the final results?

Response: we have added additional tests assessing the effect of gender and age. However, this won't affect final results. We have revised our discussion accordingly. 

  1. From table 1, demographic data is too simple, further details data is required. And Figure is more intuitive for data presentation, such as PGSI score.

Response: we have added education and employment status in demographic data and also added figure 1 for better elaboration of data.

  1. Further discussions for the differences and similarities between the present results and other studies are supposed to be added.

Response: we have added additional discussion as suggested.

  1. Line 290 and 293:The format of references in the manuscript is not uniform.

Response: we have corrected this.

Reviewer 2 Report

In the present study, the authors utilizing data previously collected for a New Zealand (NZ) national gambling survey, explored the associations of social connectedness and leisure activities with risky gambling behavior and quality of life. Poorer social connectedness and leisure activities were associated with increased gambling risk and poorer quality of life. Furthermore, social connectedness and leisure activities strongly predicted the type of gambling activities and quality of life. Moreover, Māori (NZ's indigenous population) had lower social connectedness, fewer leisure activities, a greater gambling risk, and higher psychological distress than the NZ European/Other population. These findings indicate that the risk of progressing from recreational to risky gambling is relatively higher for Māori and that social connectedness and leisure activities could contribute to this increased risk. Therefore, the authors conclude that social connectedness and leisure activities are seriously considered in public health and treatment efforts to reduce gambling harm to vulnerable populations. 

The study is well done and is the interest to a specialized audience. 

Author Response

Thank you for your kind comments! 

Round 2

Reviewer 1 Report

The author has completed some modifications. HOWEVER,  Eligible Figure Legends, readers can accurately understand the meaning of FIGURE 1 without reading the MANUSCRIPT, the author needs further modification.

Author Response

Eligible Figure Legends, readers can accurately understand the meaning of FIGURE 1 without reading the MANUSCRIPT, the author needs further modification.

Response: We have revised the figure legend as suggested, and the revised text is pasted below. 

Figure 1. Mean and 90% confidence interval of gambling risk, social connectedness, leisure activities, psychological distress, and quality of life among groups

"
